# Nanoparticle-Based RNAi Therapeutics Targeting Cancer Stem Cells: Update and Prospective

**DOI:** 10.3390/pharmaceutics13122116

**Published:** 2021-12-08

**Authors:** Yongquan Tang, Yan Chen, Zhe Zhang, Bo Tang, Zongguang Zhou, Haining Chen

**Affiliations:** 1Department of Pediatric Surgery, West China Hospital, Sichuan University, Chengdu 610041, China; yqtang@scu.edu.cn; 2Department of Gastrointestinal Surgery, West China Hospital, Sichuan University, Chengdu 610041, China; 3State Key Laboratory of Biotherapy and Cancer Center, West China Hospital, Sichuan University, Chengdu 610041, China; yanchen0524@163.com (Y.C.); scuzz@stu.scu.edu.cn (Z.Z.); 4Department of Urology, Institute of Urology, West China Hospital, Sichuan University, Chengdu 610041, China; Btanguro@scu.edu.cn

**Keywords:** nanoparticle, RNAi therapeutics, cancer stem cells, drug resistance

## Abstract

Cancer stem cells (CSCs) are characterized by intrinsic self-renewal and tumorigenic properties, and play important roles in tumor initiation, progression, and resistance to diverse forms of anticancer therapy. Accordingly, targeting signaling pathways that are critical for CSC maintenance and biofunctions, including the Wnt, Notch, Hippo, and Hedgehog signaling cascades, remains a promising therapeutic strategy in multiple cancer types. Furthermore, advances in various cancer omics approaches have largely increased our knowledge of the molecular basis of CSCs, and provided numerous novel targets for anticancer therapy. However, the majority of recently identified targets remain ‘undruggable’ through small-molecule agents, whereas the implications of exogenous RNA interference (RNAi, including siRNA and miRNA) may make it possible to translate our knowledge into therapeutics in a timely manner. With the recent advances of nanomedicine, in vivo delivery of RNAi using elaborate nanoparticles can potently overcome the intrinsic limitations of RNAi alone, as it is rapidly degraded and has unpredictable off-target side effects. Herein, we present an update on the development of RNAi-delivering nanoplatforms in CSC-targeted anticancer therapy and discuss their potential implications in clinical trials.

## 1. Introduction

Cancer is one of the leading causes of poor quality of life and mortality worldwide [1,2]. Currently, most cancer patients have micro- or macroscopic systemic metastases when they are initially diagnosed [1]. As such, systemic therapies, including chemotherapy, targeted therapy, and immunotherapy continue to be the main lines of treatment in antitumor strategies. Although numerous new drugs have emerged, drug resistance frequently occurs and remains a dominant obstacle to cancer treatment [3]. Initially, the combined administration of agents with distinct mechanisms of action was employed to solve the resistance of single-agent therapy. This approach, named polychemotherapy, is effective in the early stage of chemotherapy, while its efficacy plateaus over the following period. Multiple mechanisms of drug resistance to structurally and mechanistically distinct antitumor agents has emerged as a new challenge [4]. Most patients who die from cancer eventually develop resistance to multiple therapeutic modalities [5]. Although new therapeutic strategies, including targeted therapies and immunotherapy, have been proposed [6,7], cancer resistance continues to emerge by similar mechanisms [8,9,10]. As a result, several approaches aimed at combating unique pathways of drug resistance emerged fast and contributed significantly to improved prognosis [11].

It has been well established that most malignant tumors are composed of multiple phenotypically distinct subpopulations of cancer cells that promote varied responses to chemotherapy [12]. Among these, the cancer stem cell (CSC) concept posits the presence of a minor subpopulation that is highly capable of self-renewal and multidirectional differentiation and largely contributes to the formation of heterogeneous tumor masses [13]. The introduction of the CSC concept provides a framework for the understanding of intratumoral heterogeneity [14,15]. In recent years, drug resistance has been well-documented to be tightly linked with CSC phenotypes [14,16]. For example, breast cancer was found to have multiple signaling pathways that drive CSC properties along with chemoresistance [17,18,19,20]. These multiple drugs resisting CSCs lead to residual cancer cells during chemotherapy, which are deemed to be responsible for the subsequent relapse of tumors [16,21].

Nevertheless, CSC-targeted therapy has remained unsatisfactory in recent decades, with the primary issue being that most CSC prominences are unresponsive to small molecular agents [22]. One of the most documented mechanisms of CSC multidrug resistance (MDR) is the increased expression of plasma membrane located ATP-binding cassette (ABC) transporters that enhance drug efflux [21], and variants in these genes may impact individualized treatment efficacy or adverse drug reactions [23,24]. In addition, the CSC phenotype exhibits increased resistance to chemotherapy, not only by raising drug efflux but also by regulating many other stem characteristics, including upregulated antiapoptotic proteins and increased DNA damage repair ability [25]. In this context, several signaling pathways and malignant molecules are critical in stemness maintenance, such as the Wnt and Notch pathways, and the NF-E2-Related Factor 2 (NRF2), CD44, prominin 1 (PROM1 or CD133), epithelial cell adhesion molecule (EpCAM) and Twist family BHLH transcription factors [13,21,25]. As the synthesis or screening of chemical agents targeting protein structure and biofunctions presents several hurdles, few chemical agents have been found or constructed to effectively inhibit CSC drivers or effectors [26]. In contrast, genetic silencing by RNA interference (RNAi), including small interfering RNA (siRNA) and microRNA (miRNA), can be rapidly translated into implications. Moreover, it has been documented that the genetic silence of these CSC regulators resulted in decreased cell stemness and improved drug sensitivity in vitro [27,28,29]. However, there are numerous physical and physiological barriers preventing RNAi delivery from systemic administration to the tumor location in vivo, such as degradation by RNase, off-target effects before efficient loading of tumor tissue, and poor membrane permeability and intracellular retain [30,31]. Importantly, nanomedicines present a novel and promising therapeutic approach to overcome these limitations of conventional RNAi therapeutics [32,33]. The elaborate nanoparticles provide a physical barrier that prevents contact and elimination by RNase, and are equipped with a lipophilic outer surface that enhances transmembrane uptake. In addition, RNAi-delivering nanoparticles can be embellished with hyaluronic acids (HA) or other ligands that mediate tumor targeting. In this review, we will explore recent advances in recent decades and present a perspective on the potential targets and novel nanoplatforms that were developed to target CSCs.

## 2. CSC Modulators and Potential Targets for RNAi Therapy

CSCs were indicated to originate from nonmalignant stem or progenitor cells in several studies. The signaling pathways that are crucial for normal stem cell homeostasis and function, including the Wnt, Notch, Hedgehog and Hippo pathways, are also commonly altered and have key regulatory functions supporting the stemness maintenance, survival and drug resistance of CSCs [13,34]. Accordingly, inhibition of these signaling pathways might be a promising approach for CSC-directed therapy across multiple cancer types, which contributes to inhibition of cancer relapse and enhances the efficacy of oncological treatments. Referring to the knowledge of CSC biology, a few CSC-directed agents have been developed and even entered clinical trials [16,35]. Furthermore, some drugs currently used for the treatment of general tumors might also have a certain degree of effectiveness against CSCs, such as inhibitors of AKT and ALK in cancer therapy [36,37]. Nevertheless, the wide crosslinks and compensatory mechanisms among these pathways lead to huge limitations in therapy effectiveness. In contrast, genetic therapy using RNAi initiatives expands the number of candidates that can be targeted, achieving highly specific, widespread and possibly curative therapeutic effects [38].

### 2.1. Targeting Wnt Pathway with RNAi Therapeutics

The canonical Wnt pathway is mediated by the activation of a β-catenin (encoded by *CTNNB1*)-centered transcriptional complex with the assistance of several transcriptional cofactors [39]. Currently, at least 19 members were found in the WNT family, a series of secretory glycoproteins functioning as ligands. On the other hand, at least 10 isoforms of Frizzled family proteins act as surface receptors together with their various coreceptors, such as low-density lipoprotein receptor-related protein 5 (LRP5) and LRP6. This ligand-receptor interaction disrupts β-catenin from its degradation complex, leading to β-catenin accumulation. Subsequently, β-catenin transposes into the nucleus thus causing the activation of the T cell-specific transcription factor & lymphoid enhancer-binding factor (TCF-LEF) transcriptional complex. The activated TCF-LEF complex results in the regulation of the expression of diverse genes, especially the genes supporting CSC properties, such as *MYCN*, Cyclin D1 (*CCND1*), and *CD44* [13] (Figure 1). Cancer outcomes have a close relationship with the regulation of the Wnt pathway, including postoperational local relapses and metastasis. For example, Wnt signaling is frequently upregulated and tightly linked with poor prognosis in cancers, commonly due to inactivating mutations of adenomatous polyposis coli protein (*APC*) that mediate the ubiquitination and degradation of β-catenin [40,41]. Besides, induction of Dickkopf-related protein 1 (DKK1), an endogenous inhibitor of LRP5 and LRP6, results in the delay of cancer progression [42,43].

As a result, small-molecule inhibitors and antibodies targeting these critical components of the Wnt pathway were rapidly developed, but have not yet demonstrated sufficient effectiveness or are still being investigated in small-scale clinical trials. In a phase I clinical study, for example, PRI-724, an inhibitor of the β-catenin interaction with the transcriptional coactivator cyclic AMP response element-binding protein (CBP), converted eight patients (40%) into stable disease with a median progression-free survival (PFS) of two months [44]. Another antagonist of the β-catenin-CBP complex, E7386, significantly attenuated Wnt signaling in patient-derived hepatocellular carcinoma (HCC) xenograft models, and is being tested in early phase clinical studies [45]. Besides, antagonistic antibodies of Frizzled receptors, such as Vantictumab and Lpafricept, and anti-ROR1 antibody Cirmtuzumab are still in phase I trials without convincing outcomes [46,47,48,49,50,51].

To date, many attempts have been made for interfering Wnt signaling with RNAi in vitro and in vivo (Table 1). Treatment with nanoparticle-delivered siWNT1 as a single therapy or part of combinatorial immunotherapies acted to halt tumor growth in a lung adenocarcinoma model [52]. However, in this study, DOPC liposomes loaded with siWNT1 caused no more than half reduction of the WNT1 mRNA amount, which suggests that both siWNT1 sequences and their nanocarriers can be modified, thus improving intracellular accumulation and the efficacy of genes silencing. Several miRNAs have been identified as WNT signaling inhibitors in various cancer types, which are mainly directed to β-catenin, WNTs and WNT ligand secretion mediator (WLS) (Figure 1). For example, the miR-34 was identified to directly target multiple genes involved in the Wnt pathway, including WNT1, WNT3, LRP6, CTNNB1, and LEF1, and has a variety of functions in tumor suppression [53]. In the same way, another study found that miR-145 tends to suppress Wnt signaling through targeting CTNNB1 and significantly inhibits colon cancer cell growth [54]. In addition, some upstream activators of the Wnt pathway were also demonstrated to be modulated by miRNAs with roles in antitumor effects, such as miR-8, which directly targets the WLS [55], and miR-9, which modulates the translation of C-X-C motif chemokine receptor 4 (CXCR4) [56]. Accordingly, RNAi therapeutics targeting Wnt signaling might provide promising approaches for CSC therapy. These Wnt interfering miRNAs can be delivered by specifically designed nanoparticles, but their implication in vivo requires more evidence. Moreover, a number of genes in the downstream Wnt pathways can be targeted through RNAi delivery, especially CD44, which is also known as an identity biomarker of CSCs across multiple cancer types [57]. CD44, as a cell-surface glycoprotein, is overexpressed in several types of CSCs and frequently characterized with alternative spliced variants [58]. CD44 is primarily known as a receptor of hyaluronic acid (HA), and is also reported to bind to other extracellular matrix (ECM) ligands, including matrix metalloproteinases (MMPs), osteopontin and collagens, which are deemed to mediate intercellular interactions, ECM adhesion and migration [59,60]. The pre-mRNAs for this gene undergo complex alternative splicing to produce various lengths of variants, resulting in a range of protein isoforms with distinct biofunctions. The different functional roles of these CD44 isoforms are not fully understood. however, the isoform 4 with all the variable exons spliced was indicated to have the strongest correlation with CSC properties in various cancer types, such as breast cancer [61,62], colorectal cancer [63,64,65], liver cancer [66], and bladder cancer [67]. The HA binds to and activates CD44 signaling pathways that induce enhanced cell proliferation and survival, and modulates the cytoskeleton to promote cellular motility. Mounting evidence has demonstrated that a subpopulation of cancer cells with positive CD44 and negative or low expression CD24 (CD44^+^/CD24^−/low^) are characterized by high tumorigenicity, as a few hundred of these cells were able to form solid tumors that was found to regain their parental heterogeneity into NOD/SCID mice [68]. Moreover, CD44 was also well known to be critical in stemness maintenance in various cancers, including breast cancer [61,69,70], liver cancer [71], pancreatic cancer [72] and bladder cancer [73]. Decreased CSC phenotypes were found by interfering with CD44 expression in these cancer types. As a result, several RNAi-delivered nanoparticles were designed for cancer therapy through silencing CD44 individually or combined with other antitumor drugs [74,75]. These separate studies indicated that nanoparticle implication significantly increase the work efficacy of RNAi-mediated CD44 knockdown in vivo, meanwhile, which may be further improved through the modification of these nanoparticles.

### 2.2. Targeting Notch Pathway with RNAi Therapeutics

Similar to the Wnt pathway, the Notch signaling pathway is another developing pathway that mediates intercellular communication, and has a great correlation to multiple aspects of cancer biology, especially CSC properties and tumor immunity [95,96]. This pathway functions through transmembrane ligands and receptors interaction, which comprise Delta-like ligand 1 (DLL1), DLL3 and DLL4, Jagged 1 (JAG1) and JAG2 as canonical Notch ligands, and Notch 1–4 paralogues as Notch receptors [97]. This interaction between neighboring cells induces a two-step proteolytic cleavage of the Notch receptor, with the first-step cleavage performed by disintegrin and metalloproteinase domain-containing protein (ADAM) enzymes, either ADAM10 or ADAM17, and the second-step cleavage mediated by γ-secretase. Furthermore, the cleaved Notch intracellular domain (NICD) is released and translocates into the nucleus to regulate the expression of a range of genes, especially CSC-correlated genes such as *MYC*, *CCND3* and *ERBB2*, where it is combined with several other transcriptional cofactors, such as Mastermind- like 1 (MAML1), (Figure 2) [98,99,100]. Notably, the significance of the Notch signaling outputs in the context of CSCs is highlighted by the findings that Notch signaling interference has the potential to simultaneously repress tumorigenesis and drug resistance. Several Notch-pathway inhibitors with distinct targets and mechanisms have been developed or are now under clinical investigation. The γ-secretase is the first target used for designing inhibitors of Notch signaling. Inhibition of γ-secretase halts NICD release by blocking the second cleavage of Notch receptors, which was shown to have strong antitumor activity in various preclinical cancer models, such as pancreatic adenocarcinoma (PDAC) and T cell acute lymphoblastic leukemia (T-ALL) [101,102]. However, the majority of these γ-secretase inhibitors have been discontinued, most commonly owing to the unfavorable outcomes in phase I/II clinical studies [13]. In addition, their off-tumor side effects are another frequent problem typically involving the gastrointestinal system and electrolyte balance. In addition to small-molecule agents, several antagonistic monoclonal antibodies (mAbs) have been developed to target distinct domains of Notch ligands and receptors, which is another strategy to inhibit aberrant Notch signaling. For the most investigated example, Demcizumab is a humanized anti-DLL4 IgG2 mAb, whose antitumor efficacy in combination with specific first-line antitumor drugs has been tested on PDAC and non-small cell lung cancer (NSCLC) respectively in various phase I/II trials [103,104]. Brontictuzumab is designed as an antagonistic mAb of NOTCH1 mAb; however, it was revealed to have limited antitumor activity in several clinical studies focusing on hematological malignancies and solid tumors [105,106]. Similar to γ-secretase inhibitors, most clinical studies of these mAbs were suspended due to their unfavorable results in early phase studies.

As such, targeting Notch components with RNAi therapeutics modified with nanomedicine might be another promising strategy to inhibit aberrant Notch signaling in cancer treatment (Table 1). CSCs are considered to largely contribute to tumor relapse in hepatocellular carcinoma (HCC), accounting for poor survival, while a micellar nanoparticle that delivers siNOTCH1 was able to efficiently suppress NOTCH1 expression in HCC cells, leading to increased sensitivity to platinum and decreased CSC percentage in a xenograft model [54]. Several other studies demonstrated the feasibility of inhibiting Notch signaling by delivering siRNA targeting Notch ligands or receptors in vivo [107,108,109,110]. For example, siNOTCH1-loaded nanoparticles significantly inhibit Notch signaling, thereby attenuating rheumatoid arthritis in mouse models [107,110]. Thus, nanoparticle-aided highly effective siRNA showed promising implications in Notch-directed cancer therapy. In addition to siRNA, several miRNAs involved in the regulation of Notch signaling were used as monotherapy or codelivery with chemical drugs in various preclinical cancer models (Figure 2), whereas mounting miRNAs are involved in the regulation of the Notch signaling and cancer stemness [111]. Most notably, miR-34a was well documented in targeting and attenuating the expression of NOTCH1 and led to progression arrest in multiple cancer types [47,112]. Several subsequent studies showed that nanoparticle-carried miR-34a potently decreased the expression of NOTCH1, resulting in inhibition of cell proliferation and migration of breast cancer [48,49,51], and viability reduction of fibrosarcoma [51]. In CSC enriched glioma, exogenous miR-10b exposure led to suppression of NOTCH1, thus diminishing the invasiveness, angiogenesis and tumor growth in the brain, and significantly prolonging the survival of tumor-bearing mice [46]. Thus, with the assistance of nanoparticles, these siRNAs and miRNAs could be potent nucleic acid therapeutics of CSCs by interfering with Notch signaling.

### 2.3. Hippo Pathway and Potential RNAi Targets

The highly conserved Hippo signaling pathway acts to regulate the balance of cell proliferation and apoptosis [113]. The functions of the canonical Hippo signaling pathway are mediated by the transcriptional complex with coactivators Yes-associated protein 1 (YAP1) and WW domain containing transcription regulator 1 (WWTR1, usually known as TAZ), which promote the transcription of target genes involved in CSC properties, such as epithelial-to-mesenchymal transition (EMT), anti-apoptosis, and self-renewal [113]. Indeed, increased activity in YAP1 and/or TAZ led to the expansion of CSC populations and cancer progression [114]. On the other hand, the Hippo pathway is regulated by the successive activation of two kinase complexes, with the first comprising macrophage stimulating 1 (MST1) and MST2, and the second comprising large tumor suppressor kinase 1 (LATS1) and LATS2, together with the adaptors salvador family WW domain containing protein 1 (SAV1) and MOB kinase activator 1 (MOB1), respectively. In this context, the YAP1 and TAZ can be phosphorylated and driven into degradation upon the upstream signals, intercellular contact, G protein-coupled receptors and cell adhesion (Figure 3).

Targeting CSCs through blocking Hippo signaling has been well documented and has showed promising results (Table 1) [115,116]. For example, several small-molecule inhibitors of the YAP1 transcriptional complex, including Verteporfin, CA3 and vestigial-like protein 4 (VGLL4) mimicking peptide, were shown to have potent antitumor activity in various cancer types, especially inhibiting tumorigenesis, CSCs enrichment and resistance to radiation [117,118,119]. In addition, treatment with a NEDD8-activating enzyme (NAE) inhibitor leads to rapid degradation of the YAP1/TAZ complex via suppressing the activity of the cullin-Ring subtype of ubiquitin ligases that stabilize the LATS kinase complex [120,121]. Accordingly, Hippo cascades also have great potential as RNAi targets, and some attempts have been made in this area. Nanoparticle-delivered siRNAs for MST1/2 were shown to effectively suppress the expression of MST1 and MST2, and to enhance Hippo signaling thereby leading to hepatocyte proliferation [122]. In addition, several miRNAs were found to be involved in Hippo signaling regulation, and some showed great antitumor activity as therapeutics (Figure 3). The miR-195 was identified to suppress Hippo signaling by binding to the 3′-untranslated region (3′-UTR) of the human YAP1 mRNA, whose expression was validated in a separate cohort of colorectal carcinoma (CRC) and significantly associated with poor survival of patients [55]. Subsequent experiments indicated that overexpression of miR-195-5p in CRC cell lines repressed cell growth, colony formation, invasion, and migration [55]. Another study revealed that the expression of miR-582 decreased the proportion of phosphorylated YAP1/TAZ in NSCLC cells, potentially by targeting actin regulators [56]. As such, these miRNAs are potential candidates as therapeutics targeting CSCs, although this needs further investigation. However, Hippo-directed RNAi therapeutics are currently being investigated *in vitro*, and still require more preclinical evidence before entry into clinical trials.

### 2.4. Hedgehog Pathway and Potential RNAi Targets

The Hedgehog signaling pathway has an important role in embryonic development and its aberrant activity has been linked to a variety of tumor types [123]. The Hedgehog signaling is mediated by three mature hedgehog ligands, including Sonic hedgehog (SHH), Indian hedgehog (IHH) and Desert hedgehog (DHH). The binding of Hedgehog ligands to Patched (PTCH) transmembrane receptors relieves their inhibitory effect on Smoothened (SMO), thereby leading to nuclear localization and activation of GLI transcription factors. The activated GLIs drive gene expression with roles involved in cell self-renewal, proliferation, and survival (Figure 4) [123]. This pathway provides a novel target for cancer therapy because the modulation of Hedgehog signaling is tightly correlated with CSC properties [124]. The investigation of small-molecule agents targeting the Hedgehog pathway in cancer continues to be an active research area, which is mainly directed to Hedgehog ligands, SMO or GLIs. Several SMO inhibitors, including Vismodegib, Sonidegib and Glasdegib, have been approved successively since their potent activity in repressing Hedgehog signaling and cancer progression [125,126,127]. Accordingly, these targets can also be silenced with siRNA, resulting in suppression of Hedgehog signaling, which has been tested in a range of preclinical studies (Table 1) [128,129,130]. Additionally, several miRNAs have been identified to be involved in the regulation of Hedgehog signaling (Figure 4), of which some might be used as antitumor therapeutics with the assistance of nanomedicine. The upregulated miR-326 was revealed to decrease SMO expression, resulting in an elevated rate of apoptosis in chronic myeloid leukemia (CML) cells, which could be beneficial in eradicating CD34+ CML stem cells [78]. Another SMO-directed miRNA, miR-14, was found to suppress Hedgehog signaling activity through screening the 3′ untranslated regions (3′UTRs) of the Hedgehog pathway genes against a genome-wide miRNA library, which functions by cotargeting PTCH and SMO [131]. Various GLI-directed miRNAs have also been identified in different studies. For example, separate studies found that miR-378a-3p directly targets Gli3 in activated hepatic stellate cells and leads to reduced expression of Gli3 [132,133]. Research results from another group indicated that upregulated miR-324-5p significantly inhibited GLI1 expression resulting in reduced stem cell compartment, cell growth and survival in multiple myeloma [77]. In lung adenocarcinoma cells, interference with miR-182-5p mimicked GLI2 silencing and resulted in the suppression of tumorigenesis and cisplatin resistance [80]. In addition, there are several miRNAs that have been indicated to inhibit Hedgehog signaling with unclear mechanisms, such as miR-186 and miR-338-5p [79,81]. The identification of these Hedgehog pathway-directed miRNAs provides possibilities for their implications in cancer therapy as mono-delivered or co-delivered with anti-tumor drugs using nanoplatforms, while their efficacy in vivo for targeting Hedgehog signaling requires additional research.

### 2.5. Other CSCs Targets for RNAi Therapy

In addition to CD44, another well-documented CSC marker is CD133, also known as prominin 1 (PROM1), which functions to suppress stem cell differentiation. CD133 was first identified in tumor initial cells (TICs) of glioma, since injection of as few as one hundred CD133+ glioma cells produced a new mass with similar phenotypes to the original tumor, whereas injection of one hundred thousand CD133-glioma cells could not even produce a tumor. Afterwards, CD133 was validated to be a CSC marker in HCC, colorectal carcinoma, and ovarian cancers. However, the pathophysiological mechanisms of CD133 in cancer stemness maintenance remain unknown. The finding on CSC lines showed that CSCs in the G1/G0 phase have reduced CD133 activity compared to those in the G2/M phase, suggesting a tight link to the cell cycle of CD133 [134]. In addition, it was suggested that CD133 may play a function in cellular glucose metabolism. In this context, high glucose stimulation induced the upregulation of CD133 with concomitant downregulation of its phosphorylation [135]. As a result, silencing CD33 through nanoparticle-delivered RNAi is deemed to be a promising method for CSC-targeted cancer therapy. Several other CSC biomarkers & effectors, such as TWIST1, ALDH, EpCAM, glucose, and transporters (GLUTs), were also used as CSC targets for cancer therapy, which largely depend on the specific cancer types. Moreover, the number of ABC transporters was found to be correlated with the maturation state that the most primitive cells exhibit the greatest efflux activity. For example, subfamily B member 1 (ABCB1), also known as MDR-1 or p-glycoprotein (P-gp), was first identified and cloned, and was subsequently shown to be responsible for clinical MDR in many cancers, such as colorectal cancer, breast cancer, lung cancer, etc.. Afterwards, C subfamily member 1 (ABCC1) and G subfamily member 2 (ABCG2) were identified successively and found to mediate clinical MDR across cancer types. There are 48 members of the human ABC family, with some exhibiting exceptional pharmacological specificity. The most well-known reactive oxidative species (ROS) scavenger NRF2 is shown to be highly expressed in CSCs, and NRF2 silencing returns the high levels of ROS and the sensitivity to chemotherapy. A wide spectrum of agents exert antitumor activity through the production of excessive ROS, but the CSCs have an enhanced ROS elimination system to reduce ROS-mediated DNA damage and cell apoptosis. Mounting exploratory work was carried out, but no highly specific and efficient compounds targeting these CSC factors were found or synthesized. As such, numerous studies have been performed to verify the feasibility of small-molecule RNAi delivery for the abrogation of CSC-associated factors, which should be largely progressed as the rapid advance of nano-material.

## 3. Nanoplatforms for RNAi Delivery

As discussed above, RNAi-based therapy has recently come to be utilized as a novel attractive strategy for cancer treatment. However, RNAi technology presents many limitations in its potential clinical applications, including rapid clearance by the renal system, target tissue uptake selectivity, the efficiency of cellular uptake, and long-term efficacy. To overcome these obstacles, researchers have introduced the use of nonviral carriers for the delivery of RNAi molecules [136,137]. Nanocarriers as a type of nonviral carrier have attracted more considerable attention, which is capable of promoting drug administration and drug accumulation in tumor tissues through elaborate drug encapsulation, thereby maximizing therapeutic efficacy and minimizing the undesirable side effects. Thus, nanocarriers are emerging as an outstanding delivery system for RNAi molecules [137,138,139]. The extensively investigated nanocarriers applied to RNAi molecule delivery can be generally classified into four major groups: polymer-based nanoparticles, lipid-based nanoparticles, inorganic nanoparticles and bio-inspired nanoparticles (Figure 5) [140,141,142,143]. To deepen our understanding of the potential of these various nanoparticles in the delivery of RNAi molecules for cancer therapy, the following section briefly reviews the different nanocarriers for delivered RNAi molecules and the recent progress of RNAi-based therapy using nanocarriers for targeting cancer stem cells.

### 3.1. Lipid-Based Nanoparticles

Generally, lipid-based nanoparticles are artificially manufactured drug delivery vehicles in which the inner core is completely covered by an outer lipid bilayer coating. Lipid-based nanoparticles are widely used for satisfying biocompatibility, good stability, controlled drug release and targeting properties. Furthermore, for successful drug delivery, the physicochemical parameters of lipid-based nanoparticles can be modified by changing the lipid components, drug-lipid ratio, and fabrication process. In recent decades, a wide variety of lipid-based nanoparticles have been reported, including solid lipid nanoparticles, liposomes, micelles, and emulsions [144,145,146]. Among these various lipid-based nanoparticles, liposomes are the most commonly used because of their excellent performance indicators such as high stability, good bioavailability, controlled release, low toxicity, long-term circulation, and tumor-targeted specificity. Liposomes have been reported to be used as drug delivery vehicles that have attracted increasing interest in both the basic and clinical biomedical sciences. According to their surface charge distribution, liposomes are divided into three types: cationic liposomes, neutral liposomes, and anionic liposomes. Cationic liposomes are the most broadly used as RNAi delivery carriers because of their high affinity for negatively charged nucleic acids. The lipids of cationic liposomes are made up of cationic lipids and neutral auxiliary lipids, cationic lipids include DOTMA, DOTAP, DOSPA, DMRIE, and DC-Chol; and neutral auxiliary lipids include DOPE, DOPC, PE, phosphatidylcholine, and cholesterol. With the rapid progress of liposome-based technology, liposomes have evolved to multifunctional pharmaceutical nanocarriers combing several specific properties, such as long-circulating liposomes, pH-sensitive liposomes, and targeted liposomes [147,148,149,150]. Currently, solid lipid nanoparticles have also been applied for the systemic delivery of RNAi because they can be disinfected and lyophilized owing to their exceptional stability in humans [151]. Fortunately, these recent advances have improved the use of lipid-based nanoparticles for gene-based therapy for targeting cancer stem cells. For example, Li and colleagues developed novel GLI1-targeted siRNA nanoparticles that are functionally modified by a DSPE-HA conjugate, as a specific ligand of the CD44 receptor. In this study, the GLI1-targeted siRNA nanoparticles selectively eliminated gastric CSCs for dual-targeting CD44 and GLI1, and consequently exhibited impressive therapeutic efficacy in gastric cancer [74]. A separate study demonstrated that the drug resistance of hepatocellular carcinoma can be overcome through eliminating HCC CSCs by codelivering Bmi1 siRNA with cisplatin in cationic nanoparticles [152].

### 3.2. Polymer-Based Nanoparticles

Polymer-based nanoparticles are well-exploited carriers for RNAi delivery. In terms of origin, they are classified into two major groups: natural and synthetic polymer-based nanoparticles [153,154]. The natural polymers used for gene-based therapy include chitosan, atelocollagen, folate (FA), HA, and gelatin, and are biocompatible, biodegradable, and generally nontoxic, even at high concentrations. Chitosan has been successfully used in gene delivery systems [155,156,157]. Novel chitosan nanoparticles were developed to deliver functional miRNA mimics to macrophages through regulating ABCA1 expression and cholesterol efflux to target atherosclerotic lesions [158]. Apart from natural polymer-based nanoparticles, synthetic polymer-based nanoparticles have also been used in the delivery of RNAi molecules. Synthetic polymer-based nanoparticles dominate the majority of gene delivery systems, which mainly consist of chitosan derivatives, PLGA, PEI, PVA, PLA, PEG, and PAMAM, etc. Similar to natural polymers, synthetic polymers are characterized by good stability, high drug-loading capacity, and biodegradability [159,160,161]. Noteworthily, synthetic polymers are relatively effortless to be modified with ligand bindings and stimuli-responsive units for controlled release and targeted delivery. However, some synthetic polymers could not be directly utilized for RNAi molecule delivery owing to a lack of cationic motifs, thereby leading to low electrostatic interactions between polymers and RNAi molecules. To resolve this issue, nanoparticles need to be modified with various cationic motifs or cationic polymers [162]. The 6 (G6) TEA-core PAMAM dendrimer forms stable dendriplexes that were synthesized with a p^70S6K^ siRNA, and showed significant tumor suppression by inhibiting stemness and metastasis of ovarian cancer [163]. To effectively enhance the therapy of ovarian cancer, an impressive delivery system was designed, which includes a PPI dendrimer, a synthetic analog of LHRH peptide, paclitaxel, and siRNA molecules targeted to CD44 mRNA, together to be a specific CD44+ ovarian cancer cell death inducer. Consequently, treatment with the designed nanoparticles led to efficient ovarian cancer suppression [75]. In addition, a novel aptamer-PEI-siRNA nanoparticle was utilized for targeting the putative cancer stem cell marker EpCAM, leading to inhibition of the cancer cell proliferation. Another group demonstrated that NPsiPLK1with LY364947 pretreatment cooperatively promotes remarkable antitumor effects on breast cancer [164]. In particular, a novel synthetic siRNA nanoparticle composed of a cationic oligomer (PEI1200), a hydrophilic polymer (polyethylene glycol) and a biodegradable lipid-based crosslinking moiety was developed. This nanoparticle with siMDR1 could significantly downregulate the expression of MDR1 in human colon CSCs, resulting in effectively increasing the chemosensitivity of human colon CSCs to paclitaxel [165]. Likewise, the reduction of MALAT1 by delivering targeted nanoparticles carrying MALAT1 siRNA improved the sensitivity of glioblastoma to temozolomide [166]. It is worth noting that targeting glucose uptake by systemic delivery of NPsiGLUT3, a cationic lipid-associated PFG-PLA nanoparticle that can efficiently deliver specific siRNA targeting GLUT3, is a successful strategy for inhibiting the growth of glioma cells [167]. Taken together, although substantial progress has been achieved in the field of polymer-based nanoparticles over the past decades, there remain many concerns about the ultimate fate of synthetic polymers and their degradation products.

### 3.3. Inorganic Nanoparticles

In the past few years, inorganic nanoparticles have attracted increasing attention as potential diagnostic and therapeutic applications due to their nanoscale size and unique physicochemical characteristics compared to lipid- and polymer-based nanoparticles. In particular, inorganic nanoparticles possess excellent electrical, optical and magnetic properties, making inorganic nanoparticles applicable for the imaging and ablation of malignant tissue. Numerous inorganic nanoparticles have been reported, including mesoporous silica nanomaterials (MSNs), carbon nanotubes (CNTs), quantum dots (QDs), and metal nanoparticles (e.g., iron oxide and gold nanoparticles) [168,169,170]. Among inorganic nanoparticles, MSNs are most commonly applied due to the following critical physicochemical properties: ordered porous structure, large surface area and pore volume, high tunable particle size, two functional surfaces, and good biocompatibility. MSNs are usually modified to transform into positive charge-functionalized MSNs by appropriate approaches including amination-modification, metal cations codelivered vector and coassembly cationic polymer, because unmodified MSNs often exhibit negative charges which would reduce interactions with negatively charged nucleic acids. Therefore, in addition to surface charge modification of MSNs to enhance gene loading capacity, MSNs have been modified with multiple targeting agents to achieve better applications [171,172,173]. For example, codelivery of siTWIST-MSN-HA and cisplatin showed significant advantages in targeting specificity and targeting efficacy. These nanoparticles have potential applications for overcoming clinical challenges in ovarian and other TWIST overexpressing cancers [174]. Despite the exciting progress in the development of MSN-based nanoparticles for gene delivery, there are still many challenges that need to be addressed to facilitate their further development. In particular, the benefits and disadvantages of MSN-based carriers in vivo should be systematically investigated. Carbon nanotubes exhibit specific physical properties (structural, electronic, optical, and magnetic properties) that render them innovative materials for the delivery of therapeutic molecules. They can be either composed of single-walled carbon nanotubes (SWNT) or multi-walled carbon nanotubes (MWNT). Although SWNT and MWNT have been used to form stable complexes with siRNA to silence tumor-related gene expression in tumor cells [175,176], the applications of siRNA delivery with functionalized carbon nanotubes in targeted treatment of CSCs have not yet been demonstrated. Therefore, this attractive approach based on carbon nanotubes presents a potential therapeutic strategy by targeting CSCs using RNAi delivery across multiple different tumor types. Quantum dots are fluorescent semiconductor materials. Recent advances in new approaches to QD synthesis and covering enable quantum dots to be used as ideal candidates for imaging, diagnostics, and therapeutic delivery. In the field of therapeutic delivery, QDs have been used to promote gene therapy through delivery and imaging of treatment with RNAi [177,178,179,180,181]. However, little is known about the effect of QDs/RNAi complexes in CSCs. The application of QDs/RNAi nanoparticles for gene silencing in CSCs needs to be further explored. Metal nanoparticles are another highly exploited material for inorganic nanoparticles synthesis. Gold nanoparticles (AuNPs), a group of metal nanoparticles, have been widely used in imaging, diagnostics and therapy biomedical applications. In particular, the design of AuNPs-based covalent and noncovalent RNAi nanoparticles provides a promising therapeutic option for cancer and a number of other diseases for humans [182,183]. For instance, a glucose-installed sub-50-nm unimer polyion complex-assembled gold nanoparticle (Glu-NP) was developed for systemic delivery of siRNA to GLUT1-overexpressing breast cancer stem-like cells. Subsequent results suggested that multifunctional modified gold nanoparticles could be a promising nanoparticle for CSC-targeted cancer treatment [184]. It is worth noting that the potential toxicity of metal nanoparticles needs to be carefully and precisely studied in gene therapy applications.

### 3.4. Bio-Inspired Nanoparticles

In addition to the nanoparticles briefly mentioned above, researchers have extensively exploited new bio-inspired nanoparticles for gene delivery, such as exosome-mimetic nanoparticles. Exosomes are nanosized extracellular vesicles naturally secreted by cells, whose function is triggering intercellular communication by transferring biological information between cells. However, cell-derived exosomes are relatively finite, and their purification is also difficult. Thus, the generation of exosome-mimetic nanoparticles based on the knowledge of exosome surface structure and physiology is an attractive concept for the development of future favorable nanoparticles for the delivery of RNAi therapeutics. The exosome-mimetic nanoparticles display eminent physiochemical properties compared to the exosomes that originate from cells. For example, exosome-derived siRNA against RAD51- and RAD52 could decrease fibrosarcoma cell viability and proliferation [185]. In a similar attempt, exosome-mimetic nanoplatforms were designed for targeted cancer drug delivery. Fuente et al. designed a multifunctional nanoplatform mimicking exosomes, F-EMNs loaded with therapeutic RNAs (miR145), that could efficiently transport therapeutic RNAs to targeted cells. In another study, bioengineered exosome-mimetic nanoparticles were designed to deliver chemotherapeutic drugs. The results suggested that the antitumor effect of exosome-mimetic nanoparticles Raw264.7NVDox was significantly greater than conventional chemotherapeutic-loaded nanoparticles [186,187]. Research on exosome-based cancer therapies is not limited to experimental models. Several clinical studies have been completed or remain ongoing. In a phase I study, autologous dendritic cell (DC)-derived exosomes (Dex) were directly loaded with MAGE 3 antigens and tested against metastatic melanoma [188]. All of these interesting studies suggest that exosome-based RNAi delivery systems may have advantages in anti-CSC targeted cancer therapy. Moreover, some studies have indicated that DNA/RNA-based nanoparticles, as bio-inspired nanoparticles, are suitable for drug delivery and tissue engineering [141,142,143]. RNA nanotechnology was applied to design RNA nanoparticles containing anti-miR21 and the CD133 aptamer payloads for targeting TNBC. These RNA nanoparticles displayed not only high tumor-targeting specificity but also high efficacy for tumor growth inhibition in TNBC, which further highlighted the potential application of DNA/RNA-based nanoparticles for cancer therapy [189]. Similarly, hTERT promoter-driven VISA nanoparticle-delivered miR-34a (TV-miR-34a) was utilized in BCSCs and presented a great therapeutic effect. In this context, the VISA vectors represent essentially a VP16-GAL4-WPRE integrated systemic amplifier. In brief, TV-miR-34a can significantly inhibit breast cancer cell growth, which has great application potential in breast cancer therapy [190]. Meanwhile, bio-inspired functional lipoprotein-like nanoparticles have been studied for gene delivery [191]. For example, CXCR4 receptor-stimulated lipoprotein-like nanoparticles carrying miR-34a achieved efficient accumulation in glioma initiating cells and subsequently availably restrained glioma initiating cell stemness and chemoresistance [192]. Accordingly, there are still many challenges and opportunities for bio-inspired nanoparticles, and they will certainly play a critical role in the realization of multifunctional nanoparticles for RNAi delivery.

## 4. Conclusions

The CSC hypothesis posits that CSCs are greatly responsible for tumor heterogeneity, tumorigenesis, and therapy resistance, having an important role in cancer initiation and progression [14,25]. In particular, CSCs and their fueling heterogeneous mass are widely recognized to facilitate cancer resistance to various therapy approaches, which are directly correlated with poor clinical outcomes [193,194]. In this contexts are characterized by low proliferative ability but a high rate of asymmetric divisions that produce two cell populations, with one cell population succeeding instemness and the other cell population obtaining high proliferative capacity [195]. Therefore, effective treatment strategies must focus on inactively proliferative CSCs, while most traditional therapeutics are directed to highly growing non-CSCs [3,14]. In recent years, substantial advances have been achieved in various areas of cancer gene therapy, especially with the assistance of rapidly developing delivery materials that greatly improve the stability and targeting capacity of nucleic acids in vivo. Moreover, achievements in genomics research largely increased our understanding of the genetic basis of cancers and provided a range of new targets for therapy [196]. However, the majority of recently identified targets remain ‘undruggable’ by chemical agents. As such, the potential of exogenous RNAi may make it possible to translate our knowledge into therapeutics in a timely manner. More than a decade after the initial implication of RNAi in cancer treatment, several RNAi-based therapeutics have acquired regulatory approvals to be tested in early phase clinical trials [32]. In addition, a range of new miRNAs that have potential in CSC regulation have been revealed with the progress in epigenomics studies, which provide emerging candidates as CSC-directed RNAi therapeutics. With the recent advances of nanomedicine, in vivo delivery of RNAi using elaborate nanoparticles can potently overcome the intrinsic limitations of RNAi alone being rapidly degraded and having unpredictable off-target outcomes; however, their broad application will require continued efforts, especially on the RNAi stability, interfering efficiency and targeting ability. It is important to strictly audit the performance of these recombination agents as they are considered to be prompted into later stage trials; however, a group of RNAi therapeutics directing cancer has shown promising clinical efficacy through subcutaneous administration [32]. This highlights the possibility of siRNA therapeutics in clinical applications, and suggests that RNAi has broad potential in cancer therapy in humans.

As discussed before, there are mounting siRNAs and miRNAs that are involved in CSC suppression. It should be believed that many of them can be delivered in vivo for CSC-directed therapy. In addition, a large number of nanoparticles endowed with stability and targeting capacity showed promising results as RNAi cargoes. Therefore, their distinct combinations provide mounting possibilities for potential implications for in vivo investigation.

## Figures and Tables

**Figure 1 pharmaceutics-13-02116-f001:**
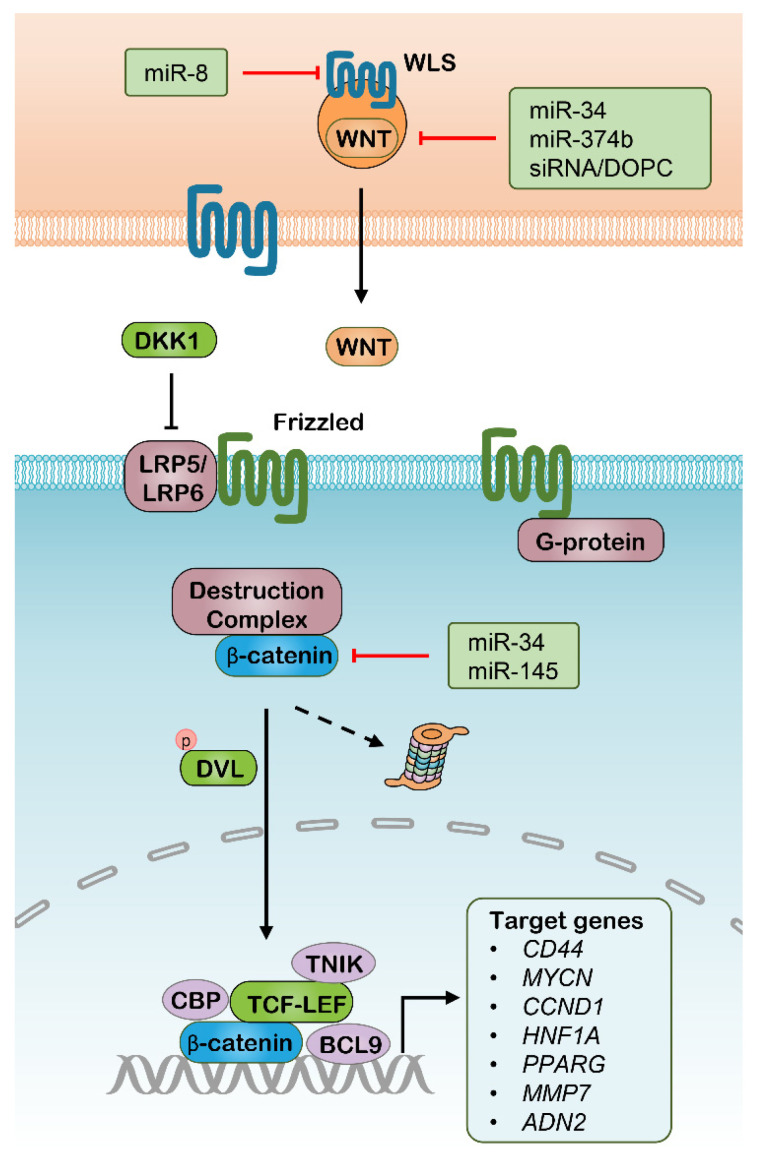
The canonical Wnt signaling pathway and potential RNAi targets. WLS, WNT ligand secretion mediator; DKK1, Dickkopf–related protein 1; LRP, Lipoprotein receptor–related protein; DVL, Disheveled Segment Polarity Protein; CBP, Cyclic AMP response element–binding protein; TNIK, TRAF2 And NCK Interacting Kinase; TCF, T cell–specific transcription factor; LEF, Lymphoid enhancer–binding factor; BCL9, B–Cell Lymphoma 9 Protein.

**Figure 2 pharmaceutics-13-02116-f002:**
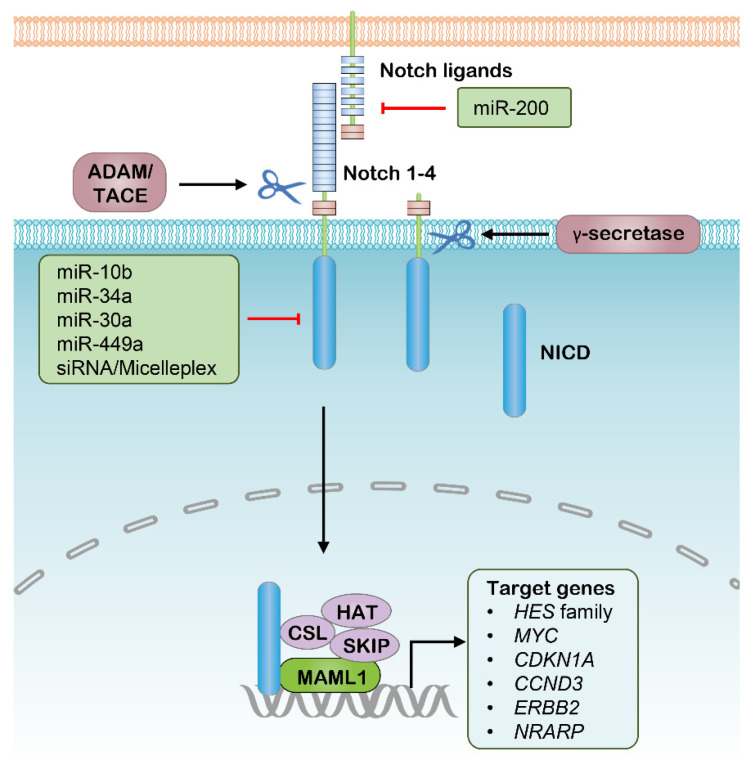
The canonical Notch signaling pathway and potential RNAi targets. ADAM, Disintegrin and metalloproteinase domain–containing protein; TACE, Tumor necrosis factor–α converting enzyme; NICD, Notch intracellular domain; CSL, CBF1/Su(H)/Lag–1; HAT, Histone acetyltransferase; SKIP, Ski–interacting protein; MAML1, Mastermind–like 1.

**Figure 3 pharmaceutics-13-02116-f003:**
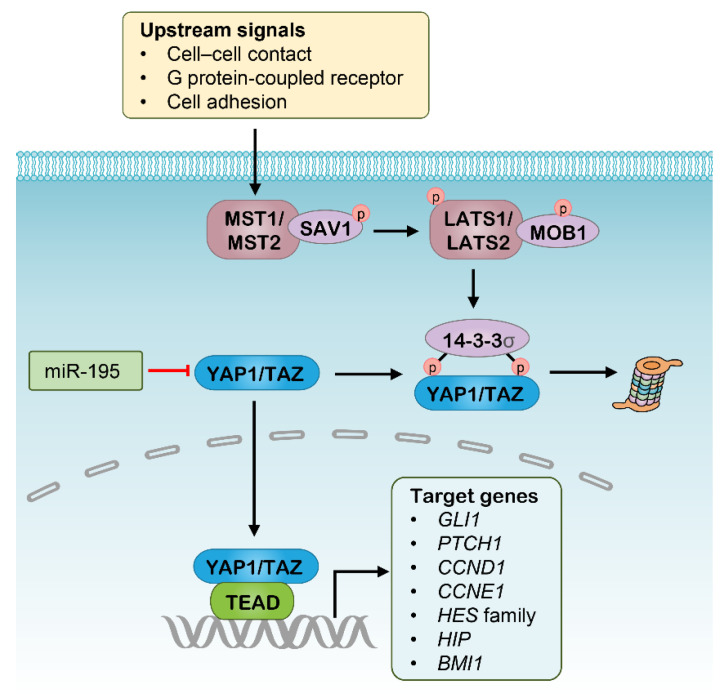
The Hippo signaling pathway and potential RNAi targets. MST, Macrophage stimulating; SAV1, Salvador homologue 1; LATS, Large tumor suppresser kinase; MOB1, MOB domain kinase activator 1; YAP1, Yes-associated protein 1; TAZ, WW domain containing transcription regulator 1; TEAD, TEA domain.

**Figure 4 pharmaceutics-13-02116-f004:**
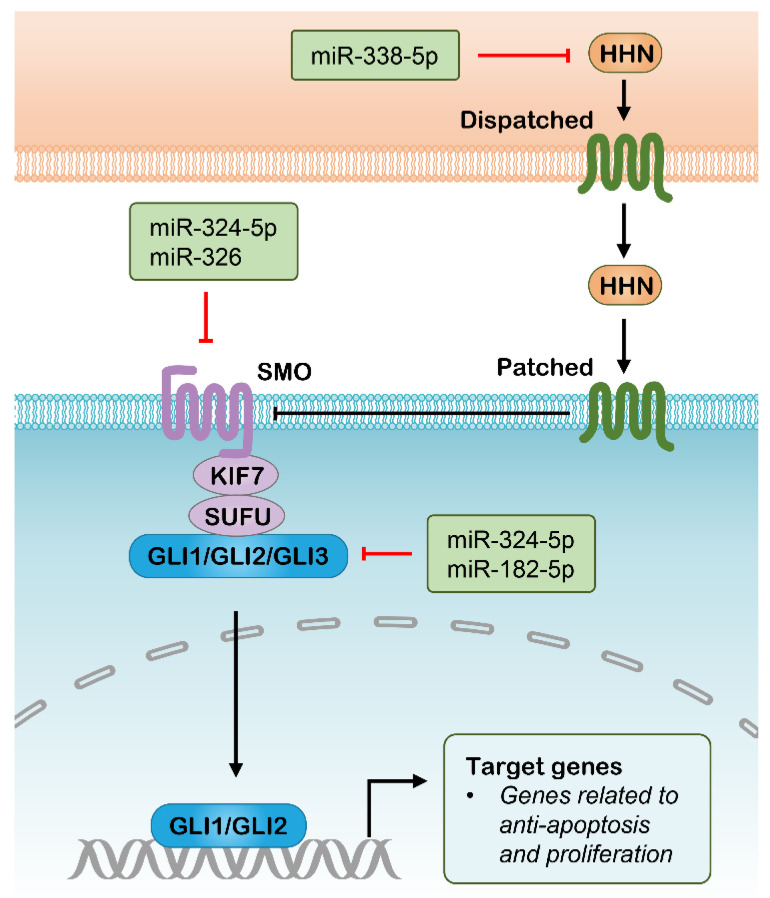
Hedgehog signaling pathway and potential RNAi targets. HHN, Hedgehog N–terminal fragment; SMO, Smoothened; KIF7, Kinesin family member 7; SUFU, suppressor of fused homologue; GLI, Glioma-Associated Oncogene Homolog 1.

**Figure 5 pharmaceutics-13-02116-f005:**
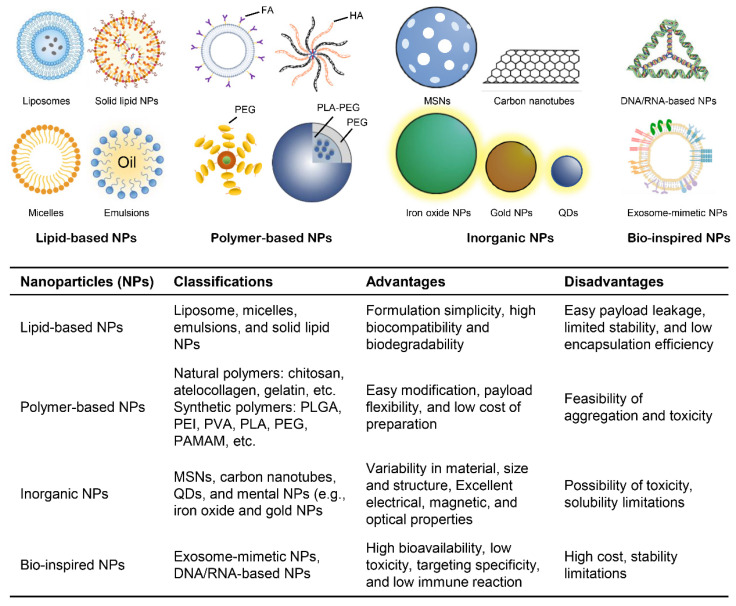
Classes of RNAi–delivered nanoparticles. FA, Folate; HA, Hyaluronic acid; PEG, Polyethylene glycol; PLA, Polylactic acid; MSNs, Mesoporous silica nanomaterials; PLGA, Polylactic–co–glycolic acid; PVA, Polyvinyl alcohol; PAMAM, Polyamidoamine; QDs, Quantum dots.

**Table 1 pharmaceutics-13-02116-t001:** Investigational RNAi therapeutics targeting CSC regulators.

RNAi	Targets	Cancer Types	Reference
Wnt-directed RNAi			
miR-8	WLS ^1^	Colorectal cancer	[55]
miR-34	WNT1 WNT3 LRP6 β-catenin LEF1	Non-small cell lung cancer Breast cancer	[53]
miR-145	β-catenin	Colon cancer	[54]
miR-9	CXCR4 ^2^	Oral squamous cell carcinoma	[56]
miR-374b	WNT16	T-cell Lymphoblastic Lymphoma	[76]
siWNT1/DOPC ^3^	WNT1	Lung adenocarcinoma	[52]
Notch-directed RNAi			
miR-200	JAG1	Prostate cancer	[77]
miR-10b	NOTCH1	Glioblastoma	[78]
miR-34a miR-34a/HA-chitosan ^4^ miR-34a/HP-IPECs ^5^ miR-34a/NPs ^6^ miR-34a/MM ^7^	NOTCH1	Colon cancer Triple-negative breast cancer Triple-negative breast cancer Triple-negative breast cancer Fibrosarcoma	[79] [80] [81] [82] [83]
miR-30a	NOTCH1~2	B- and T-cell malignancies	[84]
miR-449a	NOTCH1~2	Laryngeal cancer	[85]
siNOTCH1/Micelleplex	NOTCH1	Hepatocellular carcinoma	[86]
Hippo-directed RNAi			
miR-195	YAP1	Colorectal cancer	[87]
miR-582-5p	NCKAP1 ^8^ PIP5K1C ^9^	Non-small cell lung cancer	[88]
Hedgehog-directed RNAi			
miR-324-5p	GLI1 SMO	Medulloblastoma Multiple myeloma	[89] [90]
miR-326	SMO	Chronic myeloid leukemia	[91]
miR-186	ATAD2 ^10^	Retinoblastoma	[92]
miR-182-5p	GLI2	Lung adenocarcinoma	[93]
miR-338-5p	HHN	Glioma	[94]

^1^ Wnt ligand secretion mediator; ^2^ C-X-C motif chemokine receptor 4; ^3^ Avanti polar lipids liposome; ^4^ Hyaluronic acid-chitosan nanoparticles; ^5^ Hyaluronic acid/protamine sulfate interpolyelectrolyte complex; ^6^ Poly (lactic-co-glycolic acid) nanoparticles; ^7^ Mixed nanosized polymeric micelles; ^8^ NCK associated protein 1; ^9^ Phosphatidylinositol-4-phosphate 5-kinase type 1 γ; ^10^ ATPase family AAA domain containing 2.

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
