# Peer review of "Nanoparticle-Based RNAi Therapeutics Targeting Cancer Stem Cells: Update and Prospective"

_pharmaceutics, 2021, doi:10.3390/pharmaceutics13122116_

Round 1

Reviewer 1 Report

The article entitled “Nanoparticle-based RNAi Therapeutics Targeting Cancer Stem Cells: Update and Prospective” by Tang et al., describes the inherent self-renewal and tumorigenic capabilities of CSC play critical roles in tumor start, development, and resistance to anticancer treatment. Targeting the WNT, Notch, Hippo, and Hedgehog signaling cascades, for example, remains a viable therapeutic strategy in numerous cancer types. The authors also discuss the breakthroughs in cancer omics that have greatly expanded our understanding of CSCs' molecular underpinnings and revealed several new anti-cancer strategies. Exogenous RNA interference (RNAi, including siRNA and miRNA) may quickly become a viable therapeutic option. They also updated on RNAi-delivering nanoplatforms for CSC-targeted anticancer treatment and their potential role in clinical trials. This review is exciting and well-written.

Author Response

Dear Reviewer,

We would like to take this opportunity to express our heartfelt thanks for the precious time you spent making constructive remarks.

The article entitled “Nanoparticle-based RNAi Therapeutics Targeting Cancer Stem Cells: Update and Prospective” by Tang et al., describes the inherent self-renewal and tumorigenic capabilities of CSC play critical roles in tumor start, development, and resistance to anticancer treatment. Targeting the WNT, Notch, Hippo, and Hedgehog signaling cascades, for example, remains a viable therapeutic strategy in numerous cancer types. The authors also discuss the breakthroughs in cancer omics that have greatly expanded our understanding of CSCs' molecular underpinnings and revealed several new anti-cancer strategies. Exogenous RNA interference (RNAi, including siRNA and miRNA) may quickly become a viable therapeutic option. They also updated on RNAi-delivering nanoplatforms for CSC-targeted anticancer treatment and their potential role in clinical trials. This review is exciting and well-written.

Response: Thank you very much for your encouragement.

Reviewer 2 Report

The manuscript entitled "Nanoparticle-based RNAi Therapeutics Targeting Cancer Stem 2 Cells: Update and Prospective" by Tang and others mentioned about the Nanoparticles aided RNAi techniques in combating the cancer stem cells. The topic is most relevant to the scientific community specially cancer clinicians but novelty of the review is repetitive information. So I recommend authors to revise the manuscript after adding the key points from following DIO with adequate discussions in the each chapter

1)doi: 10.1007/s00277-019-03713-y

2)doi: 10.1002/hep4.1462

3)doi: 10.1177/2472630318816668

4)doi: 10.1016/j.apsb.2020.09.016

5)doi: 10.3390/pharmaceutics11110615

6)doi: 10.1016/j.biomaterials.2016.07.036

7)doi: 10.1016/j.jconrel.2015.05.274

Minor points

1) english editing is required

2) Authors should derive the new hypothesis from the published work so that a complete shift in the novel information available to the scientific community.

3) authors did not mention the usefulness of exosomes and following recent publications mentioned about the engineered exosomes in combating the cancer stem cell.

https://doi.org/10.3390/bioengineering8110158

Author Response

Dear Reviewer,

We would like to take this opportunity to express our heartfelt thanks to you and the anonymous reviewers for the valuable suggestions and comments. Please kindly find attached the Revised Manuscript and Point-By-Point Response.

The manuscript entitled "Nanoparticle-based RNAi Therapeutics Targeting Cancer Stem 2 Cells: Update and Prospective" by Tang and others mentioned about the Nanoparticles aided RNAi techniques in combating the cancer stem cells. The topic is most relevant to the scientific community specially cancer clinicians but novelty of the review is repetitive information. So I recommend authors to revise the manuscript after adding the key points from following DIO with adequate discussions in the each chapter, 1) doi: 10.1007/s00277-019-03713-y, 2) doi: 10.1002/hep4.1462, 3) doi: 10.1177/2472630318816668, 4) doi: 10.1016/j.apsb.2020.09.016, 5) doi: 10.3390/pharmaceutics11110615, 6) doi: 10.1016/j.biomaterials.2016.07.036, 7) doi: 10.1016/j.jconrel.2015.05.274

Minor points

1) english editing is required

2) Authors should derive the new hypothesis from the published work so that a complete shift in the novel information available to the scientific community.

3) authors did not mention the usefulness of exosomes and following recent publications mentioned about the engineered exosomes in combating the cancer stem cell.

  • English editing is required

Response: Thank you for your suggestion. The language has been checked and polished by two native English speakers independently.

  • Authors should derive the new hypothesis from the published work so that a complete shift in the novel information available to the scientific community.

Response: Thank you very much for your valuable comments. In this review, we aim to present an update on the development of RNAi-delivering nanoplatforms in CSCs-targeted anti-cancer therapy. Actually, quite a few studies present a direct investigation on CSCs therapy of RNAi-delivering nanoplatforms. Thus, as discussed in this review, most crosslinks between RNAi-nanoparticle and CSCs were derived from our comprehensive understanding. In addition, as you mentioned, we believe they should be more obvious for readers. We have added more new hypotheses, and re-written the sentences to make it more accurate.

  • Authors did not mention the usefulness of exosomes and following recent publications mentioned about the engineered exosomes in combating the cancer stem cell.

Response: Thank you for your useful suggestions. We have replenished the related content about the applications of exosomes or exosomes-mimetics nanoparticles as drug delivery vehicles for cancer therapy. However, there are rarely reports of the use of exosomes or exosomes-mimetics nanoparticles as RNAi delivery vehicles in anti-CSC targeted cancer therapy. Although the study about exosomes-based therapy in anti-CSC targeted cancer therapy is limited, the current studies remind us the potential application of exosomes-based therapy in anti-CSC targeted cancer therapy.

Reviewer 3 Report

The review presents the data on application of microRNA to treat cancer stem cells using nanoparticles. The review is excellently written and easy to read and understand. The authors covered the most of microRNA found at present time. Of course, the number of them change everyday but the authors covered the most important. Also, they described the most of modern nanoparticles developed for small RNA delivery. They and their properties are mentioned in some manner. The number of references is 183 and they give a deep insight into topic covered. The figures are very good and illustrate the topic of review. As for English, I am not native speaker and for me English is acceptable. Thus, my estimation of present manuscript is very positive. I did not find any serious remarks to mention. I think the review can be published in present form.

Author Response

Dear Reviewer,

We would like to take this opportunity to express our heartfelt thanks for the precious time you spent making constructive remarks.

The review presents the data on application of microRNA to treat cancer stem cells using nanoparticles. The review is excellently written and easy to read and understand. The authors covered the most of microRNA found at present time. Of course, the number of them change every day but the authors covered the most important. Also, they described the most of modern nanoparticles developed for small RNA delivery. They and their properties are mentioned in some manner. The number of references is 183 and they give a deep insight into topic covered. The figures are very good and illustrate the topic of review. As for English, I am not native speaker and for me English is acceptable. Thus, my estimation of present manuscript is very positive. I did not find any serious remarks to mention. I think the review can be published in present form.

Response: Thank you for your encouragement.

Reviewer 4 Report

The authors overview nanoparticle-based RNAi therapeutics intended to target cancer stems cells as anti-cancer therapy. Basically, the paper is well written and covers most of significant outcome. For this reason, I consider the paper can be appeared in Pharmaceutics. However, as this topic has numerously been published elsewhere the authors should be able to justify the novelty of the information and state the differences from previous papers. The nanoplatforms for RNAi delivery shown in Figure 5 are not actually designed for RNAi delivery (seem to be very common!). In addition, many references are cited in the previous review papers with similar topics. Prior to publication the authors need to address the comments mentioned above.

Author Response

Dear Reviewer,

We would like to take this opportunity to express our heartfelt thanks to you and the anonymous reviewers for the valuable suggestions and comments. Please kindly find attached the Revised Manuscript and Point-By-Point Response.

The authors overview nanoparticle-based RNAi therapeutics intended to target cancer stems cells as anti-cancer therapy. Basically, the paper is well written and covers most of significant outcome. For this reason, I consider the paper can be appeared in Pharmaceutics. However, as this topic has numerously been published elsewhere the authors should be able to justify the novelty of the information and state the differences from previous papers. The nanoplatforms for RNAi delivery shown in Figure 5 are not actually designed for RNAi delivery (seem to be very common!). In addition, many references are cited in the previous review papers with similar topics. Prior to publication the authors need to address the comments mentioned above.

  • As this topic has numerously been published elsewhere the authors should be able to justify the novelty of the information and state the differences from previous papers.

Response: Thank you for your critical comments. In this review, we aim to present an update on the development of RNAi-delivering nanoplatforms in CSCs-targeted anti-cancer therapy. Actually, quite few previous studies had a direct investigation on CSCs therapy of RNAi-delivering nanoplatforms. Thus, as discussed in this review, most crosslinks between RNAi-nanoparticle and CSCs are comprehensively concluded from the previous studies. We believe that they should be more obvious for readers. We also have re-written the sentence to make it more accurate.  

  • The nanoplatforms for RNAi delivery shown in Figure 5 are not actually designed for RNAi delivery (seem to be very common!).

Response: Thank you very much for your valuable comments. We have reviewed the different classes of nanoparticles. The major part of the mentioned nanoparticles has been designed for the use of RNAi delivery. Nevertheless, we mainly focus on the conception of engineering nanoparticles for RNAi delivery in anti-CSC targeted cancer therapy. Additionally, the study of nanoparticles for RNAi delivery in anti-CSC targeted cancer therapy is limited. Therefore, we have not described the designed nanoplatforms for RNAi delivery in details.

  • Many references are cited in the previous review papers with similar topics.

Response: Thank you so much for your careful check. There are some reviews talking about the implication of nanoparticle-delivering RNAi in vivo, so we have many common references. However, our review mainly focuses on CSCs-directed cancer therapy, with which there are much less papers with similar topics. We have updated many references as showed in manuscript.

Reviewer 5 Report

pharmaceutics-1464067

This paper presents a review about the development of RNAi-delivering nanoplatforms in CSCs-targeted anti-cancer therapy. Authors describe the recent advances in the past decades and present a perspective on the potential targets and novel nanoplatforms that were developed to target CSCs. They show how with the recent advances of nano- medicine, in vivo delivery of RNAi using elaborate nanoparticles can potently overcome the intrinsic limitations of RNAi alone being rapidly degraded and having unpredictable off-target outcomes. This review is well written and structured. The references are correct and the figures made by the authors themselves are clarifying and of high quality. There are only a couple of points that I think can improve the manuscript. On the one hand there are references (6, 7, 10, 27, 45, 57, 67, 78, 79, 100, 134, 170) that can be replaced by more recent ones. Although some refer to specific topics, many others have a general nature and can at least be completed with more recent ones (at least 6, 7 and 45). On the other hand, since the title corresponds to such a broad topic, it is not easy to write a review condensing all the information in a few pages. In this sense, the sections are well structured but there are some especially short, for example the one referring to inorganic nanoparticles where the gold ones are named almost in passing. I understand that it is not easy, given the extension of the field, but I think that the authors should make an effort and expand more in this section, in full swing.

Finally, although the review is quite complete in certain sections, a greater individual contribution from the authors is lacking in the final conclusions and specifically their opinon in the potential future applications. I believe that the manuscript in an improved version could reach the level of quality required in this type of revision.

Author Response

Dear reviewer,

We would like to take this opportunity to express our heartfelt thanks to you and the anonymous reviewers for the valuable suggestions and comments. Please kindly find attached the Revised Manuscript and Point-By-Point Response.

This paper presents a review about the development of RNAi-delivering nanoplatforms in CSCs-targeted anti-cancer therapy. Authors describe the recent advances in the past decades and present a perspective on the potential targets and novel nanoplatforms that were developed to target CSCs. They show how with the recent advances of nano- medicine, in vivo delivery of RNAi using elaborate nanoparticles can potently overcome the intrinsic limitations of RNAi alone being rapidly degraded and having unpredictable off-target outcomes. This review is well written and structured. The references are correct and the figures made by the authors themselves are clarifying and of high quality. There are only a couple of points that I think can improve the manuscript. On the one hand there are references (6, 7, 10, 27, 45, 57, 67, 78, 79, 100, 134, 170) that can be replaced by more recent ones. Although some refer to specific topics, many others have a general nature and can at least be completed with more recent ones (at least 6, 7 and 45). On the other hand, since the title corresponds to such a broad topic, it is not easy to write a review condensing all the information in a few pages. In this sense, the sections are well structured but there are some especially short, for example the one referring to inorganic nanoparticles where the gold ones are named almost in passing. I understand that it is not easy, given the extension of the field, but I think that the authors should make an effort and expand more in this section, in full swing.

Finally, although the review is quite complete in certain sections, a greater individual contribution from the authors is lacking in the final conclusions and specifically their opinon in the potential future applications. I believe that the manuscript in an improved version could reach the level of quality required in this type of revision.

  • There are references (6, 7, 10, 27, 45, 57, 67, 78, 79, 100, 134, 170) that can be replaced by more recent ones. Although some refer to specific topics, many others have a general nature and can at least be completed with more recent ones (at least 6, 7 and 45).

Response: Thank you for your careful check. Most references you pointed out have been updated, but the references 45 and 57 are preserved because no more recent studies provide the similar evidence we needed.

  • The sections are well structured but there are some especially short, for example the one referring to inorganic nanoparticles where the gold ones are named almost in passing. The authors should make an effort and expand more in this section, in full swing.

Response: Thank you very much for your kind suggestions. We have made an effort to expand the related content in the section of nanoplatforms for RNAi delivery. Concretely, we have added content to the section of inorganic nanoparticles and bio-inspired nanoparticles.

  • Greater individual contribution from the authors is lacking in the final conclusions and specifically their opinion in the potential future applications.

Response: We gratefully appreciate for your valuable comments. In this review, we aim to present an update on the development of RNAi-delivering nanoplatforms in CSCs-targeted anti-cancer therapy. Actually, quite a few studies present a direct investigation on CSCs therapy of RNAi-delivering nanoplatforms. Thus, as discussed in this review, most crosslinks between RNAi-nanoparticle and CSCs were derived from our comprehensive understanding. We believe that they should be more obvious for readers. We have added more new hypotheses, and re-written the sentences to make it more accurate.

Round 2

Reviewer 2 Report

The manuscript entitled "Nanoparticle-based RNAi Therapeutics Targeting Cancer Stem Cells: Update and Prospective" by Tang and others imporved considerably after the revision and Now I recommend the manuscript for publication in Phramaceutics. I congratulate the authors.